# Investigating Sources of Marine Litter and Developing Coping Strategies in Scuba Diving Spots in Taiwan

Ping-I Lin [1], Gordon Chih-Ming Ku [2], Hsiao-Hsien Lin [3], Chin-Hsien Hsu [4],*, Hung-Chih Chi [5],* and Yi-Ching Chen [5]

1   Department of Recreation and Sport Management, Shu-Te University, Kaohsiung 824, Taiwan; machiall@stu.edu.tw

2   Department of Sport Management, National Taiwan University of Sport, Taichung 404, Taiwan; gordonku@gm.ntus.edu.tw

3   School of Physical Education, Jiaying University, Meizhou 514015, China; chrishome12001@yahoo.com.tw

4   Department of Leisure Industry Management, National Chin-Yi University of Technology, Taichung 411, Taiwan

5   Department of Environmental Engineering, Da-Yeh University, Changhua 515, Taiwan; yiching@mail.dyu.edu.tw

*   Correspondence: hsu6292000@yahoo.com.tw (C.-H.H.); chikimno@gmail.com (H.-C.C.)

**Abstract:** Marine debris and floating marine debris issues have recently become a matter of great concern. The present study selected Kenting National Park and Northeast Cape and Yilan Coast National Scenic Area as the survey areas, where most of the popular scuba diving spots in Taiwan are located, to identify the volume, types, and sources of marine litter. The findings could be regarded as the foundation for future study and the suggestions for managerial strategies. The visual and line transect methods were used to conduct fourteen investigations of marine litter in four scuba diving spots from June 2020 to November 2020. Descriptive analysis and the chi-square test were used to analyze the volume, types, and sources of marine litter, as well as the different distributions under diverse locations, terrains, season, and tides. The results indicate that 2841 pieces of marine litter are identified, including 1786 (63%) plastic containers, 312 (11%) plastic bags, 254 (9%) disposable tableware for take-out beverages, 285 (10%) other materials, 72 (2%) cigarette butts, and 30 (1%) fishery and recreational fishing pieces. Different seasons, locations, and tides cause a significantly different marine litter distribution among these areas. The findings are expected to promote source reduction, develop shore and underwater cleaning proposals, and enhance marine protection education.

**Keywords:** scuba diving; marine litter; underwater line transect; visual method; action research; sustainable marine environment

## 1. Introduction

Marine environmental issues have received a great deal of attention, particularly in the areas of climate change, global warming, sea-level rise, industrial pollution, overfishing, ocean acidification, marine debris, etc. [1–5]. Specifically, the problem of a large amount of garbage affecting the ocean results from people's daily lives [5,6]. Currently, about 8 million tons of plastics enter the ocean every year, causing harm to marine life, ecosystems, human health, tourism, and the economy [7–9]. Land-based sources, as opposed to marine-based sources, are considered the dominant input of plastics into oceans [10,11].

This pollution will cause and entire biological ecosystem to be affected, and even die [10]. Ultimately, human security, life, and property will also be negatively affected [10–12], bringing morally negative perceptions to human beings.

Although Taiwan responds to the trend of marine environmental protection, according to a survey by Green Peace and The Society of Wilderness, more than 150,000 bags and 646 tons of debris have been found along Taiwan's coastline [13]. The average amount

of debris per kilometer of Taiwan's coastline is far more than that of Japan and Korea, especially along the coastline between Nanya Fishing Port and Bitou, which has been called "the dirtiest coastline in Taiwan" for two consecutive years [14]. In addition, most of the surrounding marine debris found on the northeast coast in the Ruifang District of New Taipei City is bamboo mixed with Styrofoam, fishing gear, and other plastic debris, covering an area of about 16,474 square meters. There is still a lot of room for improvement in Taiwan regarding the issue of marine environmental protection, and more in-depth investigations are needed to identify the amounts, types, and sources of debris as a basis for formulating improvement strategies [15].

The above-mentioned marine litter originate are from various human activities, and the marine litter in the coastal area is mainly produced by tourists [16–19]. Therefore, it is important to understand people's perceptions of marine litter. The types of debris that pollute the ocean vary according to season and region. According to the Ocean Conservancy advocacy group's report [20], cigarette butts were the most common type of marine debris for three consecutive years. The second most common type has changed from PET bottles to candy wrappers, while plastic products, such as PET bottles, straws, and plastic bottle caps, have always ranked third. The analysis of Taiwan's marine debris monitoring results shows that the ranking of marine debris is: No. 1, straws; No. 2 PET, bottles; No. 3, PET bottle caps; No. 4, takeaway beverage cups; and No. 5, glass bottles [21], which shows that plastic debris is the main culprit polluting Taiwan's marine environment, and it is significantly related to the consumption habits of the Taiwanese people. In terms of scuba diving investigations, the Ocean Conservation Administration in Taiwan has conducted marine litter surveys at 18 diving sites across Taiwan, according to the International Coastal Cleanup's tables classifying the marine litter. The largest percentage of marine litter is plastic bottles (32%), followed by fishing gear and iron and aluminum cans (17%). The most species-weight ratio is "fishing nets" (39%), followed by bottles (25%) and fishing gear (11%). Furthermore, when combining and analyzing the clean ocean area and the types of marine litter, fishing gear (73%) is the most common marine litter in northern Taiwan, and plastic bottles (75%) are the most common marine litter in southern Taiwan; there is no significant difference in eastern Taiwan (57%—plastic bottles and 28%—fishery floats) [21,22].

Only by gaining a better understanding of the quantity and type distribution of marine debris in various regions, controlling the source of debris, and strengthening the advocacy and education of the marine environment among the people, can we effectively address and improve the problem of debris pollution in the marine environment, and gradually achieve the goal of sustainable environmental development. Based on the above, this study selected Kenting National Park and the Northeast and Yilan Coast National Scenic Areas, with high tourist density, as the survey areas to learn the amounts, types, and sources of marine debris, and the findings can be used as the cornerstone for further research, providing feasible management suggestions for the future.

Marine litter, which was defined by the United Nations Environment Programme (UNEP) as "any persistent, manufactured, or processed solid material discarded, disposed of, or abandoned into the coastal or marine environment," has become a major marine conservation issue of global concern in recent years, as the marine plastic debris that continues to fragment after entering the natural environment is slow to biodegrade and has the most far-reaching impact [23]. Since marine debris increases as the population grows, the Ocean Conservancy advocacy group [20] estimated that more than 250 million tons of debris will flow into the ocean by 2025. Obviously, people still do not pay enough attention to the issue of marine litter.

Marine litter has caused serious damage to the living environment of seafloor organisms and poses an even great threat to the life of 693 marine species [20]; for example, discarded fishing lines entangle the fins, flippers, wings, and throats of fish, injuring seafloor creatures and endangering their lives. The research of Page et al. [24] pointed out that a large number of fur seals are entangled and killed by marine litter every year.

However, the actual number of deaths may be underestimated, as the survey did not include the number of dead animals that sank to the seafloor, meaning their carcasses were not found or counted. In addition, the impacts of marine litter on the environment and organisms include the damage caused by large debris rubbing reefs during high tide [25], plastic sheets and plastic bags suffocating seagrass beds and mangroves [26], and entangled fishing nets and fishing lines cutting corals, sponges, and anemones [27]. Thus, marine litter can be regarded as the man-made culprit that directly destroys marine environments and the ecological balance. Therefore, exploring the sources of marine litter and effectively controlling it can minimize this negative impact.

In Taiwan's coastal waters, in addition to internal land-based debris, there are marine litter sources from shipping, fishing activities, and waste drifting from other countries. According to the analysis of monitoring data in 2015, it is known that plastic products and fishing supplies are the most significant sources of debris [28] (Figure 1). The proportions of marine debris types are plastic (89.6%), glass (7.6%), metal (1.4%), and paper (1.3%). According to the use of various kinds of debris, 72.4% of the debris is related to eating behaviors (54.0%—beverage containers and straws, and 18.3%—food tableware and containers), followed by fishing (17.8%), smoking (6.3%), and other uses (3.5%) [18].

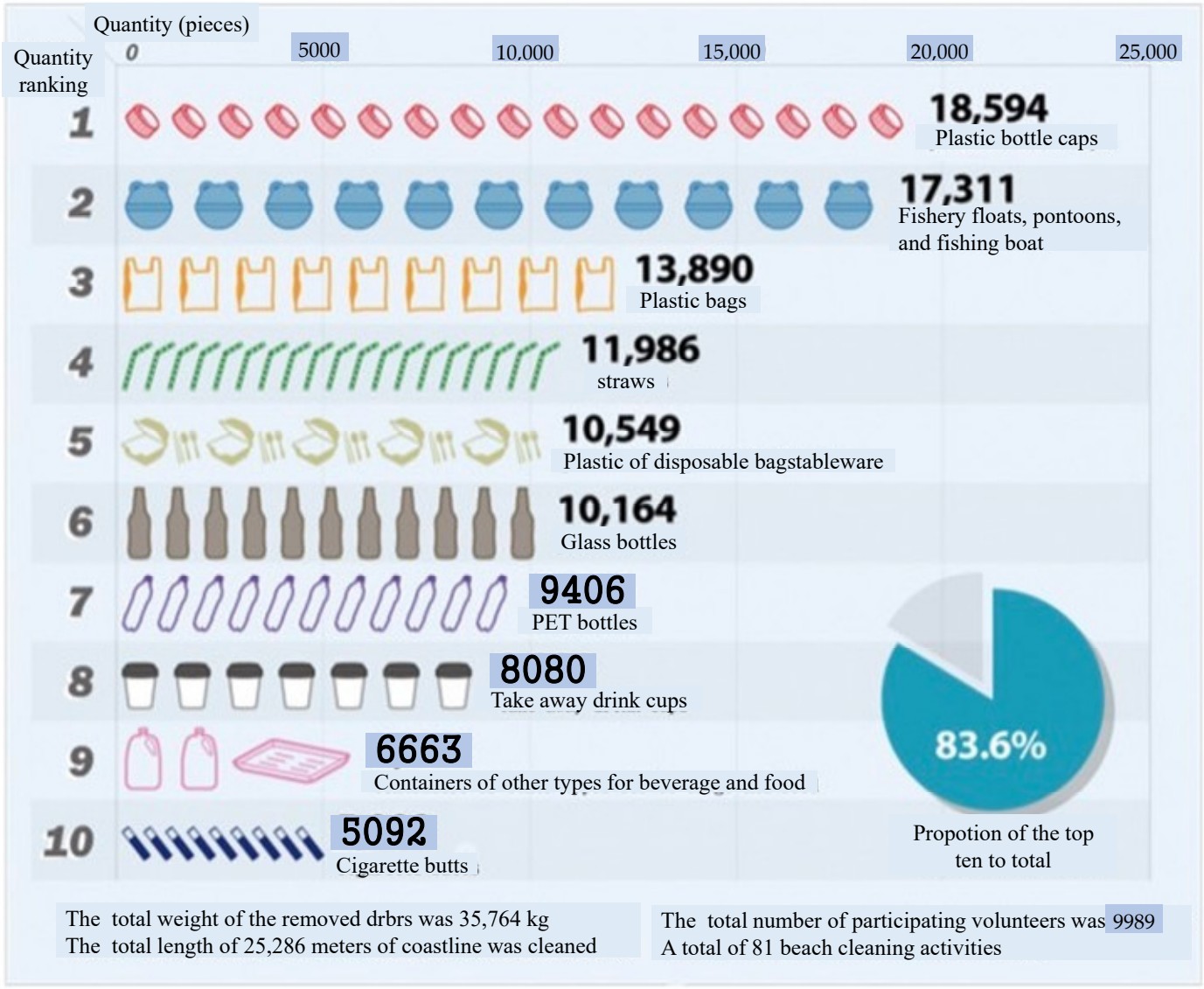

**Figure 1.** Taiwan's Civil Marine Debris Monitoring in 2015 and Proportions of Collated Data. Source: The Society of Wilderness [28].

Taiwan's marine debris mainly comes from terrestrial sources, marine sources, and foreign marine sources, and the debris differs according to the transportation routes, specific industries, or regions, with nearly 20 possible sources [15]. General debris is preliminarily estimated to account for 80% of the total amount of marine debris, while capture fisheries and marine aquaculture from marine sources account for about 15%, and the remaining 5% are wastes that migrate from other countries [28]. Therefore, Taiwan's marine debris mainly comes from the garbage produced by people's lives and is related to the living habits and consumption habits of the Taiwanese people. More importantly, as the concept of correct waste disposal and environmental pollution has not yet been implemented in daily life, it is expected that the results of this investigation on marine debris can serve as a reference for advocating education in the future.

The corresponding method for a marine debris survey can be based on three different classifications, seafloor marine debris (SMD), floating marine debris (FMD), and beach marine debris (BMD) [29]. There are six survey methods for SMD, including trawl nets, diving facilities, divers, snorkeling, sonar, and manta tow. The trawl net survey is the most commonly used survey method for SMD and some FMD (48.3%), and its universality lies in the trawl net's ability to perform fast and large-scale (including horizontal distance and vertical depth) surveys, while simultaneously conducting investigations of fish families or benthic organisms [29–32]. This study takes the survey conducted by Keller et al. (2010) on the West Coast of the United States as an example, which classified 155 sampling points as shallow, middle, and deep depths, and ranged from 55 m to 1280 m, where the product of the average net width and drag distance was used to calculate the unit area, while the quantity of marine debris collected per unit area was used as the debris density of the sampling point [31]. It was found that the deeper the location, the higher the quantity and weight of the debris. Plastic and metal waste accounted for most of the proportion, and more debris was found in the south, mainly because the area is a route for recreational ships and warships, and it is also a military disposal area. However, the limitation of this method is that trawl nets cannot be used in coral bottom areas, and trawl net mesh size will affect the size of the debris collected, meaning some debris may be left in the sea when the net is collected. The use of diving-related surveys can be considered for the water depth of tens of meters (such as snorkeling, divers, and bat tows, and the proportion of use is estimated as 41.3%). The advantage of this method is that diving can investigate all bottom materials found on the seafloor, small debris can be observed, and the impact on the environment is small, while the disadvantage is that the observed horizontal distance and vertical depth are limited, and it may be difficult to detect targets of less than 40 cm because the diving observer is towed by a small boat at a constant speed. Diving equipment and sonar are used less frequently, due to their high cost, and because neither can achieve the direct collection nor the close observation of debris; therefore, these two methods are biased towards indirect investigations. Especially with sonar surveys, only the location of the debris can be known, while debris classification cannot be completed [32,33]. Therefore, in order to overcome the above shortcomings, this study used divers to actually dive into the water for an investigation to fully understand the current situation of marine litter in the diving hotspots of Taiwan's main island.

In general, few studies in Taiwan are related to marine environmental protection, as the topics mostly focus on the current situation and policies [34], marine protection stakeholders [35], marine eco-tourism models, development strategies [36], and the analysis of the sources of marine debris [37]. Among these, only the study of Guo [37] was related to the research of marine debris; however, these studies were conducted over a decade ago. As marine litter continues uninterrupted, the value of this study is to continuously investigate the status of marine litter in Taiwan and propose targeted improvement strategies based on different times and spatial backgrounds.

## 2. Method

### 2.1. Research Design

This study used the action research method as the basis for the overall research design. In order to collect data that meets the design of this study, we employed scuba divers as the medium for data collection, adopted the underwater line transect method and the visual method, applied the waste classification table of the Ocean Conservancy advocacy group as the method of observation, and recorded the data collected in 24 surveys. The collected data were analyzed by descriptive statistics and chi-square testing, and finally, specific recommendations were made based on the analysis results. The research design is shown in Figure 2.

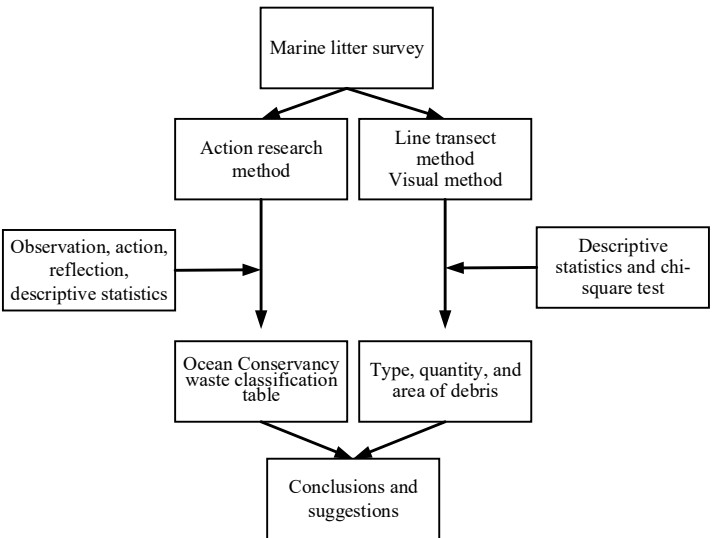

**Figure 2.** Research Design Diagram.

### 2.2. Research Sites

Taiwan is an island country located in the western Pacific; the Kuroshio current flows northeastward along eastern Taiwan year-round, and the Kuroshio Branch Current intrudes into the Taiwan Strait on the western side. The monsoon system is the driving force that changes the currents around the Taiwan Strait. In winter, the China Coastal Current is driven by the northeast monsoon entering the Taiwan Strait, and in summer, the South China Sea is affected by the southwest monsoon, entering the Taiwan Strait from the southwest [38]. This study took four diving hot spots in Kenting National Park and the Northeast Coast National Scenic Area as the research sites, namely the Maanshan Nuclear Power Plant outlet and Wanlitong Beach in Kenting, and Longdong No. 3 and Longdong No. 4 on the Northeast Coast. The Maanshan Nuclear Power Plant outlet is where the nuclear power plant discharges the cooling seawater; thus, two ocean currents of different temperatures meet to form a unique scenery with different shades of color. Therefore, a rich variety of fish gather here, and the sea area of the Maanshan Nuclear Power Plant outlet is calm year-round, making it very suitable for snorkeling and beginning diving. Wanlitong is one of the four marine ecological protection areas owned by Kenting National Park, and due to its rich terrain changes, wave erosion, and reef collapse, it is a tourist attraction for many scuba diving tourists [39]. Longdong Bay, on the Northeast Coast, is under the jurisdiction of the Northeast Coast National Scenic Area. Longdong No. 3 and Longdong No. 4 have been popular spots for scuba diving beginners in recent years due to their safe environments [40]. Moreover, because it is a popular spot for diving, the opportunities for inspection and amount of marine litter generated are worthy of observation and investigation. Thus, this study chose to conduct the survey at Wanlitong (N 21°99′56″ E 120°70′65″) [41], Maanshan Nuclear Power Plant outlet (N 21°93′21″ E

120°74′50″) [42], Longdong No. 3 (25°06′42.9″ N 121°55′01.1″ E) [43], and Longdong No. 4 (25°06′46.9″ N 121°55′11.8″ E) [44], as shown in Figure 3.

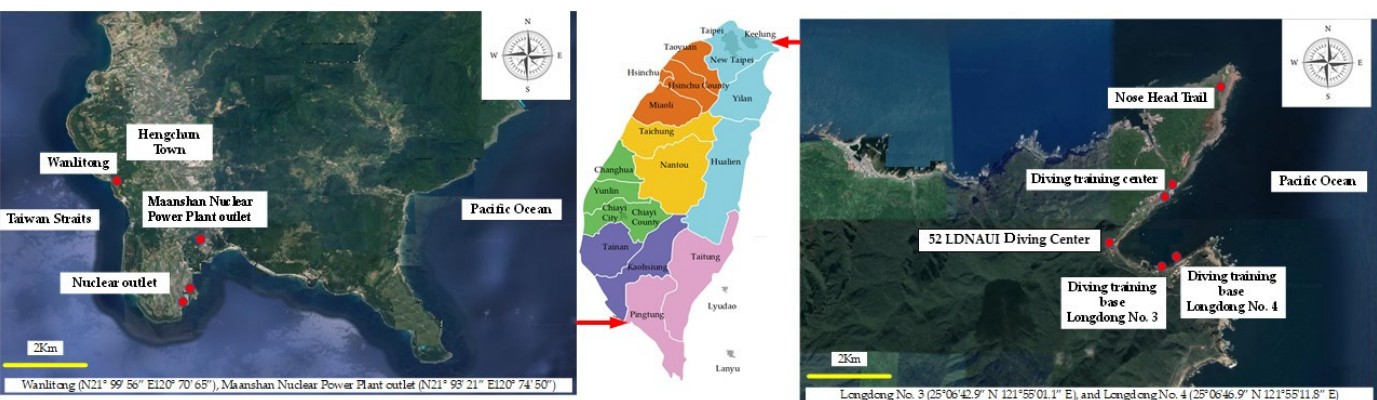

**Figure 3.** Research Sites (Wanlitong, Maanshan Nuclear Power Plant outlet, Longdong No. 3, and Longdong No. 4).

### 2.3. Action Research Method

Action research emphasizes the cyclical process of "planning, action, observation, reflection, correction, action, observation, reflection, correction" to discover potential problems and develop improvement strategies. Tsai [45] combined action and research, which are traditionally separated in action research, and advocated that practitioners should conduct research to improve their own practical work. The process of action research is the process of forming action concepts, planning, discovering facts, executing, and monitoring the results of the actions. Thus, action research is a research approach that combines action and research; "research" is to accumulate knowledge and "action" is to solve problems; "research" is to "understand", and "action" is to "improve"; action and research are the two key aspects of action research. Therefore, action research can be simply said to be "a method of systematic data collection and research, and generating actions and changes", and such research helps to generate knowledge and stimulate professional growth. Therefore, this study used action research. The researchers planned the survey actions, position, range, and weather in the scuba diving spots, and improved the survey quality using the cyclic process of action, observation, reflection, feedback, and correction [46]. The researchers in this study actually went to the above four diving hotspots (Wanlitong, Maanshan Nuclear Power Plant, Longdong No. 3, and Longdong No. 4) for scuba diving, and used the action research method to measure and collect various samples of marine litter.

### 2.4. Visual Method and Line Transect Method

The survey of this study adopted the visual method [47] and the line transect method [45] to observe the debris found in diving the hotspots. With 3 m or 5 m contours, a tape measure was used to demarcate a 100 m straight transect line for the maximum visible distance of 3 m each on the left and right sides of the line, thus, the survey range of each survey point was an underwater area of 100 × 6 m. The survey scope of the four diving areas is shown in the simulation of the underwater travel route in Figure 4. The observed data were applied to the calculation formula of marine litter density to obtain the result of marine litter density. The calculation formula is as follows:

$$D = \frac{n}{(2 \times w \times L)} \tag{1}$$

Note: D = density (pieces/square meter); $n$ = number of observations; $w$ = maximum visible distance (100 m$^2$); $L$ = length of transect line (meters).

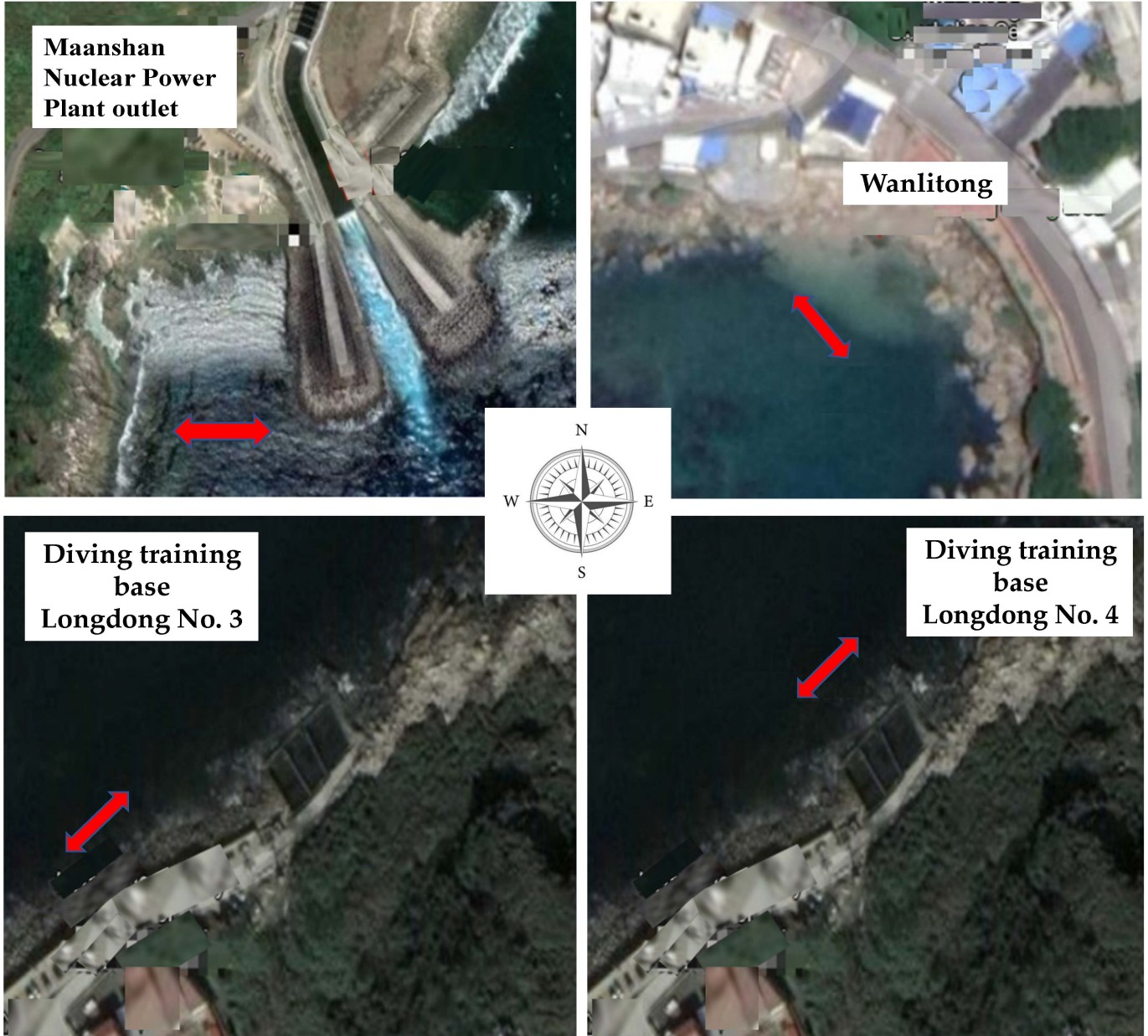

**Figure 4.** Simulation of Underwater Travel Route. Source: Google Earth Images.

*2.5. Sampling Time*

Due to various factors, such as seasons, tides, and crowds, which will affect the composition and quantity of marine debris [48,49], sampling was performed in the three seasons of spring (June to July), summer (August to September), and fall (September to November). With the aid of the 2020 tide difference table of the Central Meteorological Bureau, this study conducted two samplings at the above three popular dive sites in Southern Taiwan, namely Wanlitong, the Maanshan Nuclear Power Plant outlet, and in Northern Taiwan, namely Longdong No. 3 and No. 4, in each season, and the total number of samplings was 24. The time of each cylinder collection was about 40 min, where the first sampling was performed at low tide, the second sampling was performed after a short break on the shore, and the third was performed during high tide. In a few cases, in order to determine the amount and type of marine litter at each dive site that day, the debris classification of Ocean Conservancy was performed at the end of the two dives. It was also possible to observe whether it is easier for the debris to be situated on the seafloor

in certain submarine topographies or environments and even to understand the current density, distance, and scale when the record was taken underwater.

*2.6. Research Tools*

This study used the waste classification record table, as designed by the Ocean Conservancy advocacy group, to record statistics on marine litter. Table 1 shows the current global format, where each country uploads the information obtained after beach cleaning; thus, the annual statistics on the website can provide a better understanding of the degree of pollution caused by marine litter in various countries (Figure 5).

**Table 1.** Research Survey Timetable.

| 2020 Spring (June–July) | | | 2020 Summer (August–September) | | | 2020 Fall (September–November) | | |
|---|---|---|---|---|---|---|---|---|
| Site | Diving: 2 Cylinders Each | | Site | Diving: 2 Cylinders Each | | Site | Diving: 2 Cylinders Each | |
| A | High tide | Ebb tide | Maanshan Nuclear Power Plant outlet | High tide | Ebb tide | Maanshan Nuclear Power Plant outlet | High tide | Ebb tide |
| B | High tide | Ebb tide | Wanlitong | High tide | Ebb tide | Wanlitong | High tide | Ebb tide |
| C | High tide | Ebb tide | Longdong No. 3 | High tide | Ebb tide | Longdong No. 3 | High tide | Ebb tide |
| D | High tide | Ebb tide | Longdong No. 4 | High tide | Ebb tide | Longdong No. 4 | High tide | Ebb tide |
| Total | 8 cylinders | | Total | 8 cylinders | | Total | 8 cylinders | |

A: Maanshan Nuclear Power Plant outlet; B: Wanlitong; C: Longdong No. 3; D: Longdong No. 4.

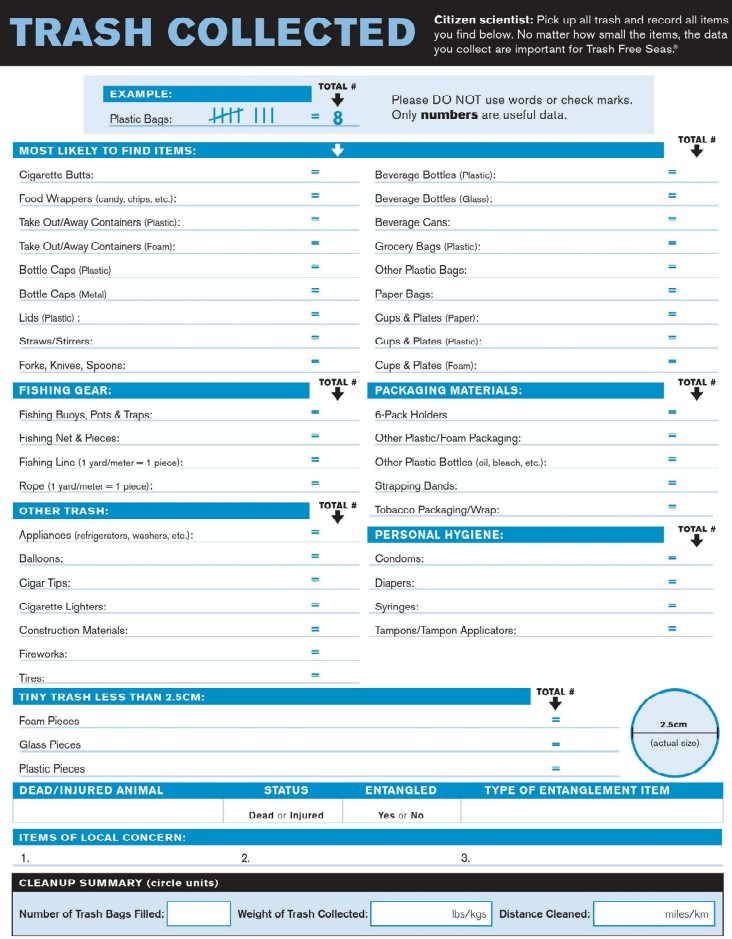

**Figure 5.** Waste Classification Table. Source: Ocean Conservancy (2022). Note: # = total value.

*2.7. Data Processing*

The data of this study were analyzed with IBM SPSS Statistics V22.0 system software, including descriptive statistics and chi-square testing. Regarding the descriptive statistics, the total amount of debris, the quantities of various types of debris, and various sources of debris were analyzed according to the location, terrain, season, and tide, and were presented in terms of frequency and percentage. Duncan's multiple range test was used to judge the difference between groups. The differences between these variables were compared by chi-square testing.

## 3. Results

*3.1. Amount and Types of Marine Litter*

A total of 2841 pieces of debris were picked up in each region, including 946 pieces (33.4%) in Wanlitong, 234 pieces (8.2%) in the Maanshan Nuclear Power Plant outlet, 959 pieces (33.7%) in Longdong No. 3, and 702 pieces (24.7%) in Longdong No. 4. There was a total of 1786 plastic containers (62.9%), 312 plastic bags (11.2%), 254 take-away beverage containers (9.1%) and disposable dishes, 285 pieces of other materials (10.1%), 72 cigarette butts (2.5%), 30 pieces of fishery and recreational fishing debris (1%), 49 personal hygiene products (1.4%), and 53 pieces of locally concerned wastes (1.8%), as shown in Table 2.

**Table 2.** Summary of the Various Types of Marine Litter.

| Area | Plastic Containers | Plastic Bags | Take-Away Beverages and Disposable Dishes | Other Materials | Cigarette Butts | Fishery and Recreational Fishing Debris | Personal Hygiene Products | Locally Concerned Wastes | Total |
|------|-----|-----|-----|-----|-----|-----|-----|-----|-----|
| B | 697 | 69 | 76 | 58 | 15 | 5 | 9 | 17 | 946 |
| A | 128 | 19 | 46 | 12 | 7 | 5 | 4 | 13 | 234 |
| C | 623 | 111 | 57 | 110 | 8 | 6 | 28 | 16 | 959 |
| D | 338 | 113 | 75 | 105 | 42 | 14 | 8 | 7 | 702 |
| Total | 1786 | 312 | 254 | 285 | 72 | 30 | 49 | 53 | 2841 |

A: Maanshan Nuclear Power Plant outlet; B: Wanlitong; C: Longdong No. 3; D: Longdong No. 4.

We found that plastic bottles are the primary source of litter in the ocean, with the same results for different sites. Furthermore, Wanlitong, Longdong No. 3, and Longdong No. 4 have more litter than the Maanshan Nuclear Power Plant outlet. While we cannot prove that the findings are affected by the season; however, we believe that this problem may be due to the seasonal effects. However, we believe that the problem may be caused by visitors who eat meals or drink bottled water or beverages without putting them in the trash can or container. In addition, plastic bottles are small and light, so ocean currents and fronts can easily push them. Wanlitong, Longdong No. 3, and Longdong No. 4 are located in areas with a low fluctuation of ocean currents and a low wind speed, easily leading to the loss of forwarding momentum when marine litter is pushed in by ocean currents or fronts, leaving the debris behind. As a result, most tourist areas have more plastic container waste than nuclear waste water outlets.

*3.2. Total and Density of Marine Litter*

The amount and density of marine litter in each area in the first season were totaled, respectively. In Wanlitong, the total quantity was 263 pieces, density = 0.219; in the Maanshan Nuclear Power Plant outlet, the total quantity was 55 pieces, density = 0.046; in Longdong No. 3, the total quantity was 283 pieces, density = 0.236; and in Longdong No. 4, the total quantity was 211 pieces, density = 0.176. The average density of the above-mentioned locations was 0.169. The quantity and density of marine litter in each area in the second season were totaled, respectively. In Wanlitong, the total quantity was 299 pieces, density = 0.249; in the Maanshan Nuclear Power Plant outlet, the total quantity was 45 pieces, density = 0.038; in Longdong No. 3, the total quantity was 331 pieces,

density = 0.276; and in Longdong No. 4, the total quantity was 278 pieces, density = 0.232. The average density of the above-mentioned locations was 0.198. The quantity and density of marine litter in each area in the third season were totaled, respectively. In Wanlitong, the total quantity was 384 pieces, density = 0.32; in the Maanshan Nuclear Power Plant outlet, the total quantity was 134 pieces, density = 0.112; in Longdong No. 3, the total quantity was 345 pieces, density = 0.288; and in Longdong No. 4, the total quantity was 213 pieces, density = 0.178. The average density of the above locations was 0.178. The details are shown in Table 3.

**Table 3.** Total Amount and Density of Marine Litter from the First Season to the Third Season.

| Season | Location | Total | Density | Average Density |
|---|---|---|---|---|
| Season 1 | B | 263 | 0.219 | 0.169 |
|  | A | 55 | 0.046 |  |
|  | C | 283 | 0.236 |  |
|  | D | 211 | 0.176 |  |
| Season 2 | B | 299 | 0.249 | 0.198 |
|  | A | 45 | 0.038 |  |
|  | C | 331 | 0.276 |  |
|  | D | 278 | 0.232 |  |
| Season 3 | B | 384 | 0.32 | 0.224 |
|  | A | 134 | 0.112 |  |
|  | C | 345 | 0.288 |  |
|  | D | 213 | 0.178 |  |

A: Maanshan Nuclear Power Plant outlet; B: Wanlitong; C: Longdong No. 3; D: Longdong No. 4 (three popular dive sites in Southern Taiwan, namely Wanlitong, the Maanshan Nuclear Power Plant outlet, and in Northern Taiwan, namely Longdong No. 3 and No. 4).

*3.3. Chi-Square Testing of Various Types of Marine Litter according to Seasons, Locations, and Tides*

The chi-square testing for seasons and types of marine litter were performed according to the two levels of the segmentation variables, respectively. The first season chi-square value = 91.158, $p$ = 0.001, which is statistically significant. The second season chi-square value = 125.459, $p$ = 0.001, which is statistically significant. The third season chi-square value = 82.860, $p$ = 0.001, which is statistically significant. Without the segmentation variable, the chi-square value = 230.942, $p$ = 0.001, which is statistically significant, as shown in Table 4.

The chi-square testing of location and type of marine litter were performed according to the two levels of the segmentation variables, respectively. The Wanlitong chi-square value = 36.668, $p$ = 0.001, which is statistically significant. The Maanshan Nuclear Power Plant outlet chi-square value = 36.183, $p$ = 0.001, which is statistically significant. The Longdong No. 3 chi-square value = 81.896, $p$ = 0.001, which is statistically significant. The Longdong No. 4 chi-square value = 11.543, $p$ = 0.643, which is not statistically significant. Without the segmentation variable, the chi-square value = 101.785, $p$ = 0.001, which is statistically significant, as shown in Table 5.

The chi-square testing of tides and types of marine litter was performed according to the two levels of the segmentation variables, respectively. The high tide chi-square value = 173.114, $p$ = 0.001, which is statistically significant. The ebb tide chi-square value = 100.025, $p$ = 0.001, which is statistically significant. Without the segmentation variable, the chi-square value = 230.942, $p$ = 0.001, which is statistically significant, as shown in Table 6.

**Table 4.** Chi-Square Testing of Marine Litter by Type and Season.

| | **Chi-Square Test** | | | |
|---|---|---|---|---|
| | **Season** | **Value** | **Degree of Freedom** | **Asymptotically Significant (Two-Tailed)** |
| Season 1 | Pearson's Chi-square | 91.158 [b] | 21 | 0.000 * |
| | Likelihood ratio | 75.128 | 21 | 0.000 * |
| | Linear-by-linear association | 10.891 | 1 | 0.001 * |
| | Number of valid observations | 812 | | |
| Season 2 | Pearson's chi-square | 125.459 [c] | 21 | 0.000 * |
| | Likelihood ratio | 128.023 | 21 | 0.000 * |
| | Linear-by-linear association | 14.719 | 1 | 0.000 * |
| | Number of valid observations | 953 | | |
| Season 3 | Pearson's chi-square | 82.860 [d] | 21 | 0.000 * |
| | Likelihood ratio | 77.376 | 21 | 0.000 * |
| | Linear-by-linear association | 7.130 | 1 | 0.008 * |
| | Number of valid observations | 1076 | | |
| Total | Pearson's chi-square | 230.942 [a] | 21 | 0.000 * |
| | Likelihood ratio | 219.602 | 21 | 0.000 * |
| | Linear-by-linear association | 33.706 | 1 | 0.000 * |
| | Number of valid observations | 2841 | | |

* $p < 0.05$. a: Maanshan Nuclear Power Plant outlet; b: Wanlitong; c: Longdong No. 3; d: Longdong No. 4 (three popular dive sites in Southern Taiwan, namely Wanlitong, the Maanshan Nuclear Power Plant outlet, and in Northern Taiwan, namely Longdong No. 3 and No. 4).

**Table 5.** Chi-square Testing of Marine Litter by Type and Location.

| | **Chi-Square Test** | | | |
|---|---|---|---|---|
| | **Location** | **Value** | **Degree of Freedom** | **Asymptotically Significant (Two-Tailed)** |
| Wanlitong | Pearson's chi-square | 36.668 | 14 | 0.001 * |
| | Likelihood ratio | 37.994 | 14 | 0.001 * |
| | Linear-by-linear association | 0.008 | 1 | 0.927 |
| | Number of valid observations | 946 | | |
| Maanshan Nuclear Power Plant outlet | Pearson's chi-square | 36.183 | 14 | 0.001 * |
| | Likelihood ratio | 31.763 | 14 | 0.004 * |
| | Linear-by-linear association | 2.943 | 1 | 0.086 |
| | Number of valid observations | 234 | | |
| Longdong No. 3 | Pearson's chi-square | 81.896 | 14 | 0.000 * |
| | Likelihood ratio | 83.887 | 14 | 0.000 * |
| | Linear-by-linear association | 9.305 | 1 | 0.002 * |
| | Number of valid observations | 959 | | |
| Longdong No. 4 | Pearson's chi-square | 11.543 | 14 | 0.643 |
| | Likelihood ratio | 11.594 | 14 | 0.639 |
| | Linear-by-linear association | 0.168 | 1 | 0.682 |
| | Number of valid observations | 702 | | |
| Total | Pearson's chi-square | 101.785 | 14 | 0.000 * |
| | Likelihood ratio | 104.313 | 14 | 0.000 * |
| | Linear-by-linear association | 7.621 | 1 | 0.006 * |
| | Number of valid observations | 2841 | | |

* $p < 0.05$. (three popular dive sites in Southern Taiwan, namely Wanlitong, the Maanshan Nuclear Power Plant outlet, and in Northern Taiwan, namely Longdong No. 3 and No. 4).

**Table 6.** Chi-Square Testing of Marine Litter by Type and Tide.

| Chi-Square Test | | | | |
|---|---|---|---|---|
| | **Tide** | **Value** | **Degree of Freedom** | **Asymptotically Significant (Two-Tailed)** |
| **High tide** | Pearson's chi-square | 173.114 [b] | 21 | 0.000 * |
| | Likelihood ratio | 154.249 | 21 | 0.000 * |
| | Linear-by-linear association | 17.149 | 1 | 0.000 * |
| | Number of valid observations | 2242 | | 0.000 * |
| **Ebb tide** | Pearson's chi-square | 100.025 [c] | 21 | 0.000 * |
| | Likelihood ratio | 107.748 | 21 | 0.000 * |
| | Linear-by-linear association | 5.588 | 1 | 0.018 * |
| | Number of valid observations | 599 | | |
| **Total** | Pearson's chi-square | 230.942 [a] | 21 | 0.000 * |
| | Likelihood ratio | 219.602 | 21 | 0.000 * |
| | Linear-by-linear association | 33.706 | 1 | 0.000 * |
| | Number of valid observations | 2841 | | |

* $p < 0.05$. a: Maanshan Nuclear Power Plant outlet; b: Wanlitong; c: Longdong No. 3; (three popular dive sites in Southern Taiwan, namely Wanlitong, the Maanshan Nuclear Power Plant outlet, and in Northern Taiwan, namely Longdong No. 3 and No. 4).

## 4. Discussion

Kenting National Park and the Northeast and Yilan Coast National Scenic Areas are tourist attractions with abundant marine resources. Due to the increasing number of tourists, local tourism activities have also increased, and the garbage produced by tourism and man-made debris affect underwater life. Guo (2013) used the transect method to investigate the amount and types of litter at six locations, including sandy shores (Baisha Bay and Jinsha Bay), rocky shores (Tsimtsilu and Longdong rock climbing sites), and fishing harbors (Danshui Second Fishing Harbor and Aodi Fishing Harbor), in order to analyze the differences in the amount and type of trash on the waterfront in terms of location, type of land, season, and tide. The results of the study showed that a total of 9319 pieces of litter were recorded in the waterfront litter section, with an average of 0.194 pieces/m$^2$. The most littered season is autumn, with an average of 0.309 pieces/m$^2$. The highest amount of litter was 0.398 pieces/m$^2$ on the rocky shore, followed by 0.149 pieces/m$^2$ on the sandy shore, and the lowest amount was 0.035 pieces/m$^2$ in the fishing harbor [49]. According to the survey results of marine litter over three seasons, Wanlitong had a total of 878 pieces, the Maanshan Nuclear Power Plant outlet had a total of 234 pieces, and Longdong No. 3 and No. 4 had 959 and 702 pieces, respectively; in terms of the total amount of debris in the three seasons, the debris generated in the sightseeing areas (Wanlitong, Longdong No. 3 and No. 4) was obviously more than that in non-tourism areas, and the types of debris were mostly plastic containers, followed by plastic bags. This study shows that the majority of the seafloor litter on the north and south coasts of Taiwan is made up of plastic waste. This result is consistent with previous studies [50–52]. Among the plastic waste, we found plastic bottles to be the most commonly seen item. Of the analysis of marine litter density, the findings are similar to the results of Watters et al. [53,54], where the average density was 3.5 pieces of marine litter per 100 m in Central California, and Zhou et al. [18], where the average density was 0.693 pieces/km$^2$ in the South China Sea [10].

This study found that the total amounts of debris in the three seasons were different, which was mainly due to the impact of the weather. On 22 August 2020, the peripheral circulation of Typhoon Bavi caused marine litter to flow from other seas to the shores of Wanlitong and the Maanshan Nuclear Power Plant outlet, which resulted in an increase in the amount of marine debris. There were also seasonal sampling factors in this study, meaning that the different water temperatures, flow directions, wind directions, and wave conditions in each season during collection affected the results of the various litter quantities and types for each season.

There were obvious differences in the types of marine litter between the different seasons. Chiu's study showed a higher density of floating marine litter in summer and fall. The higher marine litter in summer and fall might be due to monsoons. Since typhoons and afternoon thunderstorms are frequent in summer and fall in Taiwan, heavy water runoff after intense rainfall can cause litter and natural objects to flow into the ocean. This litter can move to other places along the coastline, which may be the main reason for the high litter density in summer.

Moreover, Taiwan's tourism is also divided into low and peak seasons. The number of tourists not only reflects the difference in the amount of debris, but the types of debris generated by tourists and non-tourists are also different. Furthermore, there are also different types of debris at different locations and seafloors. This study found that if the location is a tourist resort where tourists play in the water, it is easier to leave garbage when people gather in groups, and the amount that evolves into marine litter is more than that of non-tourist resorts. Finally, this study also found obvious differences in the types of marine litter according to different tides, which shows that the tide is a factor that indirectly affects the distribution of marine debris in Taiwan.

After three seasons of dive sampling and observations, we can clearly understand the threat and impact of marine litter on marine ecology. In several underwater collections, we found that the number of PET bottles and plastic products far exceeded other types of debris, As Chiu pointed out, SIMPER analysis showed regional differences in garbage composition, although plastic bottles were the most common type of litter in the waters around Taiwan. [53], including old and badly damaged PET bottles and intact plastic packaging, and some pieces even had bite marks, which suggests that seafloor creatures had tried to eat the debris by mistake. This phenomenon poses a great threat to the safety of marine life and may indirectly destroy the balance of the marine ecological environment. Countries need to institute strong and serious laws to forbid marine littering. As Choi pointed out, the reason why Korean fishermen illegally discarded their fishing gear into the sea was because of the lack of law enforcement [55,56]. Kirkley and McConnell [57,58] argue that governments are often unable to prevent the dumping of garbage into the ocean because of cost. Moreover, relative laws are often ignored. It is estimated that about 6.5 million tons of plastic waste are thrown into the ocean every year [59,60].

This study proposes several suggestions based on our findings. First, government agencies should propose supporting measures, especially for tourism areas, in order to effectively reduce the number of sources of pollutants on the seafloor; second, marine sustainability education and environmental education should be implemented in education at the grassroots level to implant the concept of sustainability into daily life; Zeppel and Ballantyne and Packer believe that formal environmental course teaching or visiting experience can convey knowledge about environmental protection and evoke correct attitudes towards environmental conservation [61,62]. Studies were done by Jensen and Schnack [63], as well as Myles and Thompson [64], also point out that to achieve a sustainable environment, knowledge learning and correct attitudes are insufficient. There is a need to change the actual behavior. Third, local residents and tourists that use the various waters should be encouraged to engage in debris recycling, and actions that prevent the garbage from entering the ocean should be thoroughly implemented; fourth, through combined physical and educational methods, residents and tourists should be invited to clean up marine litter together; fifth, tourists should be encouraged to carry their own tableware, while local stores and shops should be encouraged to refuse to use disposable tableware, and finally, a reward policy for using non-disposable tableware should be proposed.

## 5. Conclusions

The investigation of this study found that the quantity of plastic containers was the highest among all types of marine litter, most of which were PET bottles, followed by plastic bottle caps, and other beverage and food containers. The frequency of the use and demands for plastic products are extremely high, while garbage disposal methods are

very negative, which has led to the opportunity for these plastic products to flow into the ocean, causing marine environmental pollution and the frequent event of marine organisms ingesting garbage by mistake. In daily life and the process of sightseeing, people should carry their own garbage bags, and dispose of them properly, in order to avoid damaging the local marine environment ecology and destroying the habitat of seafloor animals, as well as to prevent the garbage from flowing onto the seafloor.

Since Taiwan is lacking in relevant basic research on floating marine debris and seafloor marine debris, the establishment of a marine debris database in Taiwan is relatively important, as the collection of such data helps to formulate cleanup methods. Of course, more related studies will enable this paper to receive the attention of researchers, which will allow the public and the government to face the important issue of marine environmental conservation. While plastic products bring great convenience to mankind, they also bring irreversible negative impacts to the ecology that should not be underestimated. When plastic waste enters the ecological environment, it causes damage to the ecological balance that is subsequently associated with the quality of human life. While plastic products are theoretically a problem that can be solved, a great deal of power and appropriate social concepts are required to solve the problem, meaning society can control the source and follow-up improper disposal issues through the integration of education, product design, government subsidies, and legislation.

**Author Contributions:** P.-I.L. contributions to the study included the research idea, methodology, conceptualization, funding acquisition, and writing—original draft. G.C.-M.K. contributions to the study included resources, data curation, formal analysis, founding acquisition, and the review and editing of the manuscript. Y.-C.C. and H.-H.L. contributions to the study included the investigation and project administration. C.-H.H. and H.-C.C. contributions to the study included the writing, review, and editing of the original draft. All authors have read and agreed to the published version of the manuscript.

**Funding:** This research received no external funding.

**Institutional Review Board Statement:** Not applicable. The present study does not involve humans or animals.

**Informed Consent Statement:** Not applicable. The present study does not involve humans.

**Data Availability Statement:** Not applicable.

**Conflicts of Interest:** The authors declare no conflict of interest.

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
