# Peer review of "Investigating Sources of Marine Litter and Developing Coping Strategies in Scuba Diving Spots in Taiwan"

_sustainability, doi:10.3390/su14095726_

Round 1
Reviewer 1 Report
The manuscript is organised in a part related to the data set and a part related to the suggested strategies for pollution mitigation. I think that the authors should improve the first part, providing also ancillary information on hydrodynamic as well as climatic characteristics of the sites. The observations in the Discussion section are very general, based on the differences between the sites and seasons, but these differences should be described and analysed more deeply. The authors observed that, sometimes, adverse sampling conditions (turbidity for instance) can significantly influence the data collection, therefore these cases should be removed in the statistical analysis (lines 362-368).
Is it possible to describe what are the main litter kinds that characterise the different sites, or seasons or tide level?
If the manuscript should be focused on the suggestion for mitigation actions, a larger comparison with other studies is needed.
I’m not a native English speaker, but sometimes I had difficulties to understand properly what was the meaning of the sentences, can the language be checked?
Line 17: the origin of litter was not identified in the text
Line 19: were the samplings 14? I counted 24 (4 sites for 3 seasons, with 2 tidal conditions)
Line 201: it is better to refer to water masses than to currents. It would be useful to report on the figures the main trends of currents. This can help to highlight the possible origin of litter. Line 253 (crowds): The numbers of tourists and of local citizens can be further interesting data, and the seasonal variations in peak and low touristic season. In addition, the description of the climatic features can help the discussion of lines 359-362 and 369-371, where references to the monsoon and typhoon regimes are provided but not directly correlated to the data.
What’s the range of the tide, horizontal on the beach and vertical in the water column? Is it possible to say that tidal changes can mobilise the debris previously settled on the beach?
Can you specify whether the debris collection was performed only on the sediment, or also in the water column? Can you describe what’s a cylinder collection?
Line 248: is w expressed as m2? The density would be per cubic metre. L is always 6 m, report it if you write that w is equal to 100.
Line 280: where the analyses on debris sources and terrain are? You tested the others, but not these.
Lines 264-266: can you explain better what did you mean?
In the Tables you can use codes for the different areas.
Table 1 is redundant with the text.
Lines 285-291: the reader can find the numbers in the Table, maybe in the text you can describe the trends and percentages.
Lines 294-309: please, proved the measure for density. Here again the numbers are in the Table, do not repeat them but describe the trends.
I think that figures are easier to understand than tables, in addition the information on abundance of the different kinds of detritus in each site and each season is lacking, you gave sums of the observations. This information can be interesting.
The description of the statistical analyses is redundant with the Tables. Focus on the fact that nearly all the tests indicated significant differences between the samples. This result can be easily seen on graphs (histograms, for instance).
Lines 352-355. Can you add or specify other comparisons with previous papers related to abundance and density and dealing with the kind of debris? You cited previous studies in the Introduction.
Line 359: but no relevant increase of garbage was observed in summer for that area, why?
Lines 373-377: in the site description you wrote that all the sites can be touristic. I suppose that the nuclear plant site was the less touristic one, but no detailed information was given. This can be a control for the southern area, but you have no control for the northern one. How can the reader have information on touristic or non-touristic impact and compare this to the data you collected? The role of transport by sea should be investigated.
Lines 378-379: how the tide can have a role in the differentiation of marine litter? You should mention what are the main features in the different conditions, otherwise the reader can’t understand the role of this environmental forcing.
Fig. 3. Please place a small figure of Taiwan to easily recognise the position of the two sampling areas.
Fig. 4. Provide a label with a dimensional scale (m)
Author Response
Dear Editor,
Thank you very much for forwarding to us the valuable comments from the reviewers. Their time and effort are very much appreciated. We have revised our manuscript based on their comments, and have summarized the changes in the attached file.

Reviewer 2 Report
Dear Authors,
I read your manuscript titled "Investigating Sources of Marine Litter and Developing Coping 2 Strategies in Scuba Diving Spots in Taiwan-Kenting National 3 Park and Northeast Coast National Scenic Area". In general, it is very important for the field of pollution. However, I have some suggestions. Please find below:
Introduction
- "acidification, marine debris, etc. [1-2].". Please change to acidification, marine debris, among others (References). Please consider more references to support that sentence. Some examples are:
Industrial pollution:
https://doi.org/10.1016/j.chemosphere.2020.127435
Overfishing:
https://doi.org/10.1016/j.ocecoaman.2021.105662
Debris:
https://doi.org/10.1016/j.marpolbul.2021.112821
- “plastics enter the ocean every year, causing harm to marine life, ecosystems, human health, tourism and the economy [4]”. Please include more examples: Here are some:
Ecosystems services:
http://www.scielo.org.za/scielo.php?script=sci_arttext&pid=S0038-23532020000300005
Marine life:
https://doi.org/10.1016/j.envpol.2019.112994
https://doi.org/10.1093/icesjms/fsv165
- “plastics enter the ocean every year, causing harm to marine life, ecosystems, human health, tourism and the economy [4]”. After this sentence, please include the information that many of these litter come from rivers.
https://doi.org/10.1038/ncomms15611
- “more than 150,000 bags and 646 tons of debris have been found along Taiwan’s coastline.” Please include a reference.
- “especially the coastline between Nanya Fishing Port and Bitou, which has been called “the dirtiest coastlines in Taiwan” for two consecutive years.” Please include a reference.
- “New Taipei City is bamboo mixed with styrofoam, fishing gear, and other plastic debris, covering an area of about 16,474 square meters.” Please include a reference.
- “Obviously,”. Please remove. Check it in all text and please remove.
-“The largest amount of marine litter is plastic bottles 67 (32%), the others are fishing gear and iron and aluminum cans (17%). The most species weight ratio is "fishing nets" (39%), the others are bottles (25%) and fishing gear (11%).” Reference. Is not clear to me that reference 15 support that sentence.
- “The waste produced by humans extends from the land to the seafloor, which causes a great threat to the ecological environment of the ocean.”. Please include examples of references. In fact, seems very similar with some ideas of the in first two paragraph of Introduction.
- “Taiwan’s marine debris mainly comes from terrestrial sources, marine sources, and foreign marine sources, and the debris differs according to the transportation routes, specific industries, or regions, with nearly possible sources.” Please include reference.
- “General debris is preliminarily estimated to account for 80% of the total amount of marine debris, while capture fisheries and marine aquaculture from marine sources account for about 15%, and the remaining 5% are wastes that migrate from other countries. Therefore, Taiwan’s marine debris mainly comes from the garbage produced by people’s lives and is related to the living habits and consumption habits of the Taiwanese people. More importantly, as the concept of correct waste disposal and environmental pollution has not been implemented in daily life, it is expected that the results of this investigation on marine debris can serve as a reference for advocating education in the future.” Please include reference.
- “It was found that the deeper the location, the higher the quantity and weight of the debris. Plastic and metal waste accounted for most of the proportion, and more debris was found in the south mainly because the area is a route for recreational ships and warships, and it is also a military disposal area.” Keller et al. [26]? Is not clear the author of that sentence.
General comments. Introduction may be reduced to four paragraphs. Need substantial support of references in the field (as suggested).
Method
This section seems ok. However, figures should provide reference (e.g., Google Earth?)
Line 240-245: Is not clear the author. Please include a reference if necessary.
Discussion.
- “The results of this study show the largest amount of plastic waste that is similar with previous studies [40-44].” You should reinforce the novelty of your work.
- "Thus, it was inferred that the total amount of marine litter in diving hotspots has a considerable relationship with local tourism development." Why? Please left clear this idea.
- Line 369: Please remove obvious in all text.
General comments. You should provide a depth discussion of your results showing the novelty of your work. For example, the debris come from fishing activities and their possible consequences based on literature, amog others.
Conclusion
General comments. Please reduce the size of this section.
Author Response

(The authors gave the same response as above.)

Reviewer 3 Report
sustainability-1665519
The authors should address the following comments to improve the quality of the paper:
- Lines 1-3: The title is a bit long. The details of the study areas are not necessary and can be removed. I suggest ending the title after the word “Taiwan” because, according to the abstract, there are three study areas (Kenting National Park and Northeast Cape and Yilan Coast National Scenic Area), which are too long to appear in the title
- Lines 23-24: please, provide the percent distribution of the litter composition beside the quantities
- Lines 56-73: How different is the present study from that conducted by the Ocean Conservancy advocacy group. What value is this study adding to the findings in the reports?
- Line 132: what are the shortcomings of diving facilities, divers, snorkeling, sonar, and manta tow survey methods that made me inferior to the trawl net method?
- The novelty of the study is not clear in terms of its theoretical, methodological, or practical contribution?
- Lines 219-234: please justify the choice of the research design compared to other methods.
- Lines 235-237: Also, justify selecting four study sites and those selected.
- Section 2.5: the number and timing of surveys should also be justified
- Section 2.7: Again, the appropriateness of Chi-square for data analysis and the type of Chi-square used should also be indicated.
- Section 3.3: A p-value at 99% confidence level is normally written as p= 0.001.
- Section 4: please stress the relevance of the findings to environmental sustainability, especially SDG 12 (responsible consumption and production) and SDG 14 (life under water).
- Section 5: the conclusion section should highlight the limitations of the study, and future research direction
Author Response

(The authors gave the same response as above.)

Round 2
Reviewer 2 Report
Dear Author,
Now the manuscript is ok.
Author Response
Dear Reviewer,
We rather appreciate your efforts and comments. Thank you very much!
This manuscript is a resubmission of an earlier submission. The following is a list of the peer review reports and author responses from that submission.